# Cholesteric Liquid Crystals with Thermally Stable Reflection Color from Mixtures of Completely Etherified Ethyl Cellulose Derivative and Methacrylic Acid

**DOI:** 10.3390/polym16030401

**Published:** 2024-01-31

**Authors:** Kazuma Matsumoto, Naoto Iwata, Seiichi Furumi

**Affiliations:** Department of Chemistry, Graduate School of Science, Tokyo University of Science, 1-3 Kagurazaka, Shinjuku, Tokyo 162-8601, Japan; 1322628@ed.tus.ac.jp (K.M.); n-iwata@rs.tus.ac.jp (N.I.)

**Keywords:** ethyl cellulose, Williamson ether synthesis, cholesteric liquid crystal, Bragg reflection, methacrylic acid

## Abstract

Cellulose derivatives have attracted attention as environmentally friendly materials that can exhibit a cholesteric liquid crystal (CLC) phase with visible light reflection. Previous reports have shown that the chemical structures and the degrees of substitution of cellulose derivatives have significant influence on their reflection properties. Although many studies have been reported on CLC using ethyl cellulose (EC) derivatives in which the hydroxy groups are esterified, there have been no studies on EC derivatives with etherified side chains. In this article, we optimized the Williamson ether synthesis to introduce pentyl ether groups in the EC side chain. The degree of substitution with pentyl ether group (*DS*_Pe_), confirmed via ^1^H-NMR spectroscopic measurements, was controlled using the solvent and the base concentration in this synthesis. All the etherified EC derivatives were soluble in methacrylic acid (MAA), allowing for the preparation of lyotropic CLCs with visible reflection. Although the reflection peak of lyotropic CLCs generally varies with temperature, the reflection peak of lyotropic CLCs of completely etherified EC derivatives with MAA could almost be preserved in the temperature range from 30 to 110 °C even without the aid of any crosslinking. Such thermal stability of the reflection peak of CLCs may be greatly advantageous for fabricating new photonic devices with eco-friendliness.

## 1. Introduction

Given the concerns about the mass consumption of the finite petroleum resources that remain on the Earth, the promotion of research and development on functional materials prepared from biomass resources is of prime importance for the realization of a sustainable society. Cellulose and its derivatives have recently attracted renewed interest as one of the biomass resource options because of their safety and biocompatibility. Cellulose is the naturally occurring homopolymer of *β*-D-glucopyranose, and it has been widely used as a raw material for paper dating back to the times of ancient civilizations. As one of their unique properties, cellulose derivatives can demonstrate a liquid crystal phase, allowing for dissolving them in solutions [1] or suspensions [2]. The emergence of the liquid crystal phase depends on the solubility of the polymer in the solvent and the polymer concentration. Such a liquid crystal phase is called the lyotropic liquid crystal phase. Due to these interesting properties of cellulose and its derivatives, numerous studies have been conducted to explore their application potential as functional materials [1,2].

Hydroxypropyl cellulose (HPC) and ethyl cellulose (EC), as depicted in Figure 1, are among the representative cellulose derivatives that can be obtained by reacting natural cellulose, in this case, with propylene oxide and ethylene oxide, respectively. Owing to their safety, both HPC and EC are nowadays utilized as not only ink [3,4] and pharmaceutical additives [5,6] but also thickeners and coating agents in interdisciplinary industrial fields [7]. More interestingly, EC has been reported as a biodegradable material, which fits into the concept of the circular economy [8].

Another interesting property of HPC and EC is their ability to exhibit a cholesteric liquid crystal (CLC) phase with visible light reflection in solutions [1,9,10,11,12]. When powdery HPC or EC is dissolved in solvent(s) at high concentrations, the viscous solution shows a lyotropic CLC phase with light reflection characteristics depending on the concentration of cellulose derivative. For instance, the lyotropic CLC phase appears when dissolving pristine HPC in water [1,12,13,14] or methanol [15]. As another precedent, solutions of pristine EC in organic solvents such as chloroform [11,16] or acrylic acid (AA) [17] also exhibit the lyotropic CLC phase. Such a light reflection phenomenon is regarded as a kind of Bragg reflection [18,19,20]. The mechanism of the Bragg reflection color change of HPC or EC solutions with their concentration can be explained by the difference in CLC structure. The CLC structure is characterized by periodic helical molecular assemblages. This molecular structure causes periodic modulation of the reflective index of the CLC, thereby leading to the emergence of light reflection. The maximum wavelength (*λ*) of the Bragg reflection peak is numerically expressed as the following equation:(1)λ=n·p
where *n* denotes the average reflective index of CLC and *p* is the helical pitch length [21]. The reflection peak wavelength of CLC is significantly dependent on the concentrations of HPC [13] or EC [17] solutions, leading to their wide range of potential applications as concentration indicators, reflective color displays [22], full-color recording media [23], tunable lasers [24], and so forth.

In this context, HPC can exhibit a CLC phase depending on the temperature when the hydroxy groups in its side chains are chemically modified [25]. A substance that exhibits a liquid crystal phase within a certain temperature range is called a thermotropic liquid crystal. It has been reported that when the hydroxy groups in the side chains of pristine HPC are esterified [9,25,26,27] or etherified [25], the reflection peak wavelength of CLC can be controlled by temperature in either case. Moreover, the reflection peak of CLCs from HPC derivatives generally shifts to the longer wavelength side upon undergoing a heating process. Such changes in the reflection peak wavelength can be ascribed to the modulation of *p* depending on the temperature [19]. Interestingly, the reflection properties of esterified HPC derivatives and etherified HPC derivatives differ. In the case of esterified HPC derivatives, the reflection peak appears at the longer wavelengths at the same temperature when the substitution degree of hydroxyl groups in the side chain of pristine HPC decreases [28]. However, the etherified HPC derivatives show a reflection peak at the shorter wavelength side at the same temperature, accompanied by a decline in the substitution degree [25]. Based on these precedents, it can be understood that the p value of CLC from HPC derivatives greatly depends not only on the concentration or temperature but also their substitution degree.

Previously, Gray and Guo reported that esterified EC derivatives exhibit a lyotropic CLC phase when dissolved in solvents at appropriate concentrations [11,16,18]. Like esterified or etherified HPC derivatives, the reflection peak wavelength of the solutions of esterified EC derivatives can be controlled by temperature. However, the research progress on the CLCs of EC derivatives has lagged behind that of HPC derivatives because the chemical modification of EC is not as easy as that of HPC due to the inferior solubility of pristine EC to organic solvents. In addition, etherification with alkyl halides has low reaction efficiency, whereas esterification with alkanoyl chlorides proceeds with high yields. Therefore, no reports have yet been released on CLCs prepared from etherified EC derivatives. As mentioned above, the reflection properties of CLC change significantly in the case of HPC derivatives depending on the presence or absence of carbonyl groups as well as the substitution degree. Accordingly, this situation motivates us to investigate the optical properties of etherified EC derivatives. This is extremely important to comprehend the CLC behaviors of EC derivatives for the fabrication of new cellulose-based photonic materials. Moreover, it is also essential to elucidate the effect of the degree of etherification of EC derivatives on the reflection properties of CLCs.

In this study, we optimized the etherification of hydroxy groups of pristine EC in order to obtain EC derivatives possessing pentyl ether groups (EC-Pe), as presented in Figure 1B. During the course of our systematic syntheses, we found that the degree of substitution with a pentyl ether group (*DS*_Pe_) rises with the increase in the base concentration in *N*,*N*-dimethylacetamide (DMAc) as a reaction solvent. The CLC phase appeared when dissolving the etherified EC derivatives in methacrylic acid (MAA) or AA. Such lyotropic CLCs with visible light reflection could be prepared regardless of the *DS*_Pe_ of EC derivatives only when MAA was used as the solvent. In general, lyotropic CLCs are highly susceptible to slight temperature fluctuation, so that the reflection peak wavelength is readily shifted in wide wavelength ranges by changing the temperature. Therefore, efforts have been made to tune the reflection peak wavelength by altering the temperature; however, these have involved cumbersome handling, which presents a serious problem from the technological viewpoint. Against that background, we serendipitously found a unique reflection characteristic whereby the reflection peak wavelength of CLC from completely etherified EC derivatives dissolved in MAA is almost maintained in a wide temperature range of 30–110 °C even though MAA is not polymerized. Such thermally stable Bragg reflection colors of CLCs are expected to be highly advantageous for the creation of next-generation photonic devices with eco-friendliness from cellulose.

## 2. Experimental Section

### 2.1. Materials

Three kinds of pristine EC substances with different molecular weights were purchased from Tokyo Chemical Industry (Tokyo, Japan) and used as the starting materials to synthesize EC derivatives. The viscosities of 5 wt% solutions of pristine ECs in the mixed solvent of toluene and ethanol (volume ratio 8:2), respectively, were 90–110, 45–55, and 9–11 mPa·s according to the datasheet of the manufacturer. Hereafter, each pristine EC is coded as EC*x*, where *x* is the sample number when arranged in descending order of viscosity. The number average molecular weight (*M*_n_) and weight average molecular weight (*M*_w_) of each EC were found to be 6.74 × 10^5^ and 4.29 × 10^6^ for EC1, 5.02 × 10^5^ and 2.43 × 10^6^ for EC2, and 2.58 × 10^5^ and 0.762 × 10^6^ for EC3, respectively, as determined using size exclusion chromatography (SEC) equipped with a reflective index detector (HLC-8220GPC, Tosoh, Tokyo, Japan) calibrated using the polystyrene standards. In the SEC measurements, tetrahydrofuran (THF) was employed as the eluent. The molar amount of chemically combined ethylene oxide per anhydroglucose unit, that is, the molecular substitution (*MS*), was found to be 2.50 via the ^1^H-NMR spectrum measurement of pristine EC in CDCl_3_. Therefore, the average molecular weight per anhydroglucose monomer unit could be calculated to be 232. EC was dried under vacuum at room temperature for over 24 h before use.

Dehydrated DMAc, dehydrated *N*-methyl-2-pyrrolidone (NMP), *N*,*N*-dimethylformamide (DMF), acetonitrile, and dehydrated dimethyl sulfoxide (DMSO) as solvents for the etherification of EC were obtained from Fujifilm Wako Pure Chemical Co., Inc. (Tokyo, Japan). 1-Bromopentane (PeBr), used for the synthesis of EC-Pe. MAA and AA, used as solvents of lyotropic CLCs, were purchased from Tokyo Chemical Industry (Tokyo, Japan). Sodium hydroxide (NaOH) and potassium iodide (KI), used as the catalysts of etherification, were acquired from Fujifilm Wako Pure Chemical Co., Inc. (Tokyo, Japan). All reagents except EC were used as received.

### 2.2. Syntheses of the EC Derivative Possessing Pentyl Ether Groups (EC-Pe)

EC-Pe underwent the Williamson ether synthesis in a manner similar to our previous procedures to prepare the etherified HPC derivatives [29,30]. The reaction conditions are listed in Table 1. The typical etherification procedure of EC-Pe is described as follows (Table 1, Sample code: EC1-Pe_0.15_).

In a 100 mL round-bottomed flask, 3.00 g of EC was completely dissolved in 24.0 mL of dehydrated NMP. After that, we subsequently added 4.01 mL of PeBr to the EC solution (5.00 eq. to hydroxy groups of EC). After stirring for 30 min at 65 °C, 0.72 g of powdered NaOH (0.03 g/mL for reaction solvents) and 0.27 g of KI (5.00 mol% to PeBr) were added. Subsequently, this reaction mixture was refluxed at 65 °C for 48 h. The reaction mixture was then purified via two rounds of centrifugation at 1.0 × 10^4^ rpm for 5 min to remove any sediment such as sodium bromide. Then, the supernatant was dialyzed against an equivolume mixture of methanol and water for 3 h, and the dialysis was prolonged for an additional 48 h in an equivolume mixture of THF and methanol by using a Visking dialysis tube with pore sizes of ~5 nm and a molecular weight cutoff of 1.2–1.4 × 10^4^. The product was achieved through evaporation at 35 °C in vacuo for approximately 30 min and was finally vacuum dried for a few days to obtain purified EC-Pe. The characterization of EC-Pe was carried out via FT-IR spectroscopy using an attenuated total reflection module (FT-IR4700 and ATR PRO ONE, JASCO, Tokyo, Japan), ^1^H-NMR spectroscopy (JNM-ECZ400S, JEOL, Tokyo, Japan) for the molecular structure, and SEC analysis for the *M*_n_ and *M*_w_ values.

### 2.3. Fabrication Procedure of Lyotropic CLC Cells

The EC derivatives were completely dissolved in MAA or AA using a planetary centrifugal mixer (AR-100, Thinky, Tokyo, Japan). The lyotropic CLC mixture was sandwiched between two glass substrates. The cell gap was adjusted using polytetrafluoroethylene film spacers with a thickness of ~200 µm. The edge of each cell was sealed with epoxy resin to prevent the evaporation of MAA or AA upon heating.

### 2.4. Optical Measurements of Lyotropic CLC Cells

The transmission spectrum of the lyotropic CLC cell was determined on a compact charge-coupled (CCD) spectrometer (USB2000+, Ocean Optics, Orlando, FL, USA) equipped with an optical fiber. The CLC cell was illuminated with white light from a tungsten halogen light source (Ocean Optics, HL2000). The transmitted light from the CLC cell was focused through two pieces of achromatic doublet lenses, and it was collected into the entrance of an optical fiber connected with the CCD spectrometer in a collinear arrangement with respect to both the white light source and CLC cell. The temperature of the CLC cell was precisely controlled using a hot-stage system for the optical microscope (HS82 and HS1, Mettler Toledo, Columbus, OH, USA). Polarized optical microscope (POM) images were taken with a CCD camera (EO-5012, Edmund, Barrington, NJ, USA) equipped on the microscope (IX71, Olympus, Tokyo, Japan).

### 2.5. Rheological Measurements of the Lyotropic CLCs of EC Derivatives

Viscosity measurements were conducted using a stress-controlled rheometer (MCR102, Anton Paar, Graz, Austria) equipped with a stainless-steel parallel plate with a diameter of 8 mm. The temperature was tuned by using a forced convection oven (CTD450, Anton Paar). The lyotropic CLCs were sandwiched at a gap of ~1.0 mm. The angular frequency (*ω*) dependence of the storage modulus (*G*′) and the loss modulus (*G*″) was taken on the above-mentioned rheometer. The measurements were performed in the *ω* range between 0.1 and 100 rad/s at 30 °C. The strain amplitude was adjusted in the range between 0.2 and 1.2%, which was small enough to measure the linear viscoelasticity. Prior to these measurements, the lyotropic CLCs were pre-sheared to erase any orientational history of CLC structures. The pre-shear treatment was conducted by shearing the samples at a constant shear rate of 3 s^−1^ for 350 s at 30 °C, then leaving to stand for 1 h at the same temperature after stopping the shearing force.

## 3. Results and Discussion

### 3.1. Characterization of EC-Pe

After the Williamson ether synthesis, we measured both the FT-IR and ^1^H-NMR spectra to confirm that hydroxy groups of pristine EC are substituted with pentyl ether groups. Figure 2 shows comparative FT-IR spectra between pristine EC and EC3-Pe_0.5_ (Figure 2A) and the representative ^1^H-NMR spectrum of EC3-Pe_0.5_ (Figure 2B).

As shown in Figure 2A, the FT-IR spectrum of pristine EC showed an intense peak from the O-H stretching vibration of the hydroxy groups of EC in a broad wavenumber range of 3000–3600 cm^−1^. In contrast to pristine EC, the FT-IR spectrum of the completely etherified EC derivative did not show a broad peak in the same wavenumber region. Furthermore, the intense peak from the C-H stretching vibration at 2840–3000 cm^−1^ became stronger, suggesting that etherification had proceeded. These results suggest that the hydroxy groups of pristine EC are completely substituted with pentyl ether groups. Figure 2B represents the ^1^H-NMR spectrum of EC3-Pe_0.5_ in CDCl_3_. The proton peak at 0.89 ppm can be assigned as the signal “*a*” corresponding to the terminal methyl groups in the pentyl ether groups of EC-Pe. The *DS*_Pe_ value, that is, the degree of substitution with pentyl ether group, can be quantitatively analyzed using the following equation:(2)$$DS_{\mathrm{Pe}}=A\left(7+5MS\right)/\left(3W-11A\right)$$
where *A* is the integrated value of the signal peak “*a*” and *W* is the sum of the integrated values of all protons in EC-Pe. The mathematical derivation of Equation (2) is available in Appendix B, as described below. As mentioned in Section 2.1, we adopted the *MS* value of 2.50 from the ^1^H-NMR spectrum of pristine EC. By applying the experimental results to Equation (2), the *DS*_Pe_ value of EC-Pe was estimated to be 0.50. Moreover, the values of *M*_n_ and *M*_w_ of pristine EC and EC-Pe were evaluated based on the SEC measurements using the polystyrene standards. As compiled in Table 2, both *M*_n_ and *M*_w_ of pristine EC and EC-Pe were in the same order of magnitude as those of pristine EC. This implies that no depolymerization in the main chain of EC might occur in the etherification process.

Previously, Gray and co-workers revealed that the reflection peak wavelength of CLCs from esterified EC derivatives is greatly affected by the degree of esterification, that is, the number of hydroxy groups of the EC backbone that remains [11,16,18]. These precedents motivated us to investigate the effect of the degree of etherification on optical properties even for the etherified EC derivatives. Therefore, it was necessary to find synthetic conditions that allow us to control the number of residual hydroxyl groups in the side chain of pristine EC.

### 3.2. Synthesis of Etherified EC Derivatives

A series of etherified EC derivatives were synthesized by changing the reaction conditions such as solvent or NaOH concentration, as listed in Table 1. First, we examined the effect of the solvent on the etherification of EC1 with high *M*_n_ and *M*_w_ values. Since the Williamson ether synthesis is the S_N_2 reaction, aprotic and high-polarity solvents are known to promote the reaction. Therefore, we synthesized EC-Pe using five kinds of solvents, that is, acetonitrile, DMSO, DMF, NMP, and DMAc. In all cases, the NaOH concentration in the reaction solvent was fixed at 0.03 g/mL. When acetonitrile, DMSO, and DMF were used as reaction solvents, etherification hardly proceeded because the pristine EC and NaOH were not soluble in these solvents. Alternatively, when NMP and DMAc were adopted in the Williamson ether synthesis, etherified EC derivatives could be synthesized because EC and NaOH were easily dissolved in NMP or DMAc. However, it was found that the *DS*_Pe_ can be improved from 0.15 to 0.34 by changing the solvent from NMP to DMAc (Table 1, Sample codes: EC1-Pe_0.15_ and EC1-Pe_0.34_). When considering our overall results, we concluded that DMAc is the most appropriate solvent for the etherification of EC.

Subsequently, we examined the effect of NaOH concentration in DMAc on the etherification of pristine EC. For this purpose, the NaOH concentration was increased from 0.03 g/mL to 0.05 g/mL or 0.10 g/mL. As a result, *DS*_Pe_ became 0.36 or 0.50 when the NaOH concentration in DMAc was 0.05 g/mL or 0.10 g/mL, respectively (Table 1, Sample codes: EC1-Pe_0.36_ and EC1-Pe_0.50_). These results indicated that the hydroxyl groups in the side chains of pristine EC can be completely modified by examining the base concentration. This can be attributed to the increased efficiency of the reaction between the alkoxide ion and the brominated terminal carbon of PeBr at higher base concentrations. Appendix A shows a comparison of the FT-IR spectra of pristine EC and etherified EC derivatives. The peak intensity at 3000–3600 cm^−1^, which was assigned to the O-H stretching vibration of the hydroxy groups in the EC side chain, was greatly affected by the difference in *DS*_Pe_. This peak disappeared when the base concentration in the reaction solvent was 0.10 g/mL. Therefore, it turned out that a NaOH concentration of 0.10 g/mL in DMAc enables the synthesis of a completely pentyl-etherified EC derivative with a maximum *DS*_Pe_ value of 0.50. As explained in Section 2.1, ^1^H-NMR spectrum measurement of pristine EC revealed that the molar amount of chemically combined ethylene oxide per anhydroglucose unit, that is, the *MS* value, was 2.50. Considering the chemical structure of pristine EC, the maximum *DS*_Pe_ value was 0.50 since the monomer unit of cellulose has three hydroxy groups, as depicted in Figure 1B.

Furthermore, we found that the *DS*_Pe_ value of etherified EC derivatives can be easily controlled from 0.34 to 0.50 by changing the NaOH concentration in DMAc at 0.03–0.10 g/mL. To prove the versatility of this reaction condition, we synthesized another series of EC-Pe from pristine EC with different molecular weights under the same conditions. Consequently, the completely etherified EC derivatives could be prepared regardless of the molecular weight of pristine EC (Table 1, Sample codes: EC2-Pe_0.50_, EC3-Pe_0.50_). Additionally, two kinds of EC-Pe derivatives with *DS*_Pe_ values of 0.12 and 0.29 were also synthesized from EC3 when the base concentration was adjusted to 0.02 g/mL and 0.03 g/mL, respectively (Table 1, Sample codes: EC3-Pe_0.12_, EC3-Pe_0.29_). These results highlight that the *DS*_Pe_ of EC derivatives could be easily tuned by changing the NaOH concentration.

### 3.3. Molecular Weight Dependence of EC-Pe on the Reflection Property of Its Lyotropic CLC

We found that the derivatives from pristine EC with a smaller molecular weight were suitable for preparing lyotropic CLCs with a sharp reflection peak. In this study, three kinds of completely etherified EC derivatives with different molecular weights were synthesized from EC1, EC2, and EC3 (Table 1, Sample codes: EC1-Pe_0.50_, EC2-Pe_0.50_, and EC3-Pe_0.50_). All the EC derivatives were dissolved in MAA at a polymer concentration of 65 wt%, whereupon reflection peaks appeared in the wavelength range between 420 nm and 520 nm at 30 °C for the lyotropic CLCs (Figure 3). Here, we discovered two phenomena related to the molecular weight dependence of EC-Pe on the reflection property of CLC. First, the reflection peak of CLC appeared at the shorter wavelengths with the increase in the molecular weight of EC-Pe when compared at the same temperature. The reflection peak wavelengths of the lyotropic CLCs were 517 nm for EC1-Pe_0.50_, 435 nm for EC2-Pe_0.50_, and 426 nm for EC3-Pe_0.50_. The shorter wavelength shift of the reflection peak with decreasing molecular weight suggests the decrease in the helical pitch length of CLC, corresponding to *p* in Equation (1).

As another phenomenon, the baseline in the transmission spectrum increased from ~80% to ~90%, and a sharp reflection peak emerged, accompanied by the decrease in the molecular weights of EC derivatives. To quantitatively evaluate the spectral sharpness of the reflection peaks, we focused on the half width at half maximum in the reflection peak of each lyotropic CLC. The half width at half maximum of lyotropic CLC was 103 nm for EC1-Pe_0.50_, 61 nm for EC2-Pe_0.50_, and 45 nm for EC3-Pe_0.50_. Therefore, it can be considered that the lyotropic CLC of EC3-Pe_0.50_ is most likely to be self-assembled in the molecular helical manner. Such a difference in the orientation behaviors of lyotropic CLCs can be explained by their viscosity. As shown in Appendix A, we explored the changes in the viscosities of EC1-Pe_0.50_, EC2-Pe_0.50_, and EC3-Pe_0.50_ as a function of shearing time. The measurements were conducted at a constant shear rate of 3 s^−1^ for 350 s at 30 °C. From the experimental results, the steady-state viscosity of EC-Pe in MAA was determined as follows: 1.83 kPa·s for EC1-Pe_0.50_, 1.03 kPa·s for EC2-Pe_0.50_, and 0.66 kPa·s for EC3-Pe_0.50_. These fit with our expectations as the viscosities of polymer solutions generally increase with the molecular weight. It can be considered that the lowest viscosity, of EC3-Pe_0.50_, enables the formation of highly oriented CLCs to improve their optical properties because the liquid crystal molecules are more likely to move than in the other cases. It should be noted that the low viscosity of a lyotropic CLC is very advantageous for the emergence of a vivid Bragg reflection color because air bubbles in the lyotropic CLC, which cause light scattering, can be easily removed by degassing. Therefore, we concluded that EC3 is the most suitable pristine EC for the preparation of lyotropic CLCs. From these results, the optical properties of CLCs from EC3-Pe were further investigated, and the results are presented in the following section.

### 3.4. Reflection Properties of Lyotropic CLC of EC3-Pe in MAA or AA

When lyotropic CLCs were prepared from a series of EC3-Pe with different *DS*_Pe_, we found that MAA is far more suitable than AA as a solvent to form lyotropic CLCs with visible Bragg reflection. The preparation conditions and sample definitions of lyotropic CLCs are shown in Table 3. Figure 4 shows the transmission spectra of lyotropic CLCs of a series of EC3-Pe with different *DS*_Pe_ dissolved in MAA or AA, which were determined at 30 °C. The insets of this figure also present the light reflection images.

For example, when EC3-Pe_0.12_ was completely dissolved in MAA at the concentration of 50–54 wt%, a lyotropic CLC phase with visible reflection was observed (Figure 4A, Table 3, **Samples 1**–**3**). It was found that the Bragg reflection peak wavelength can be tuned by changing the concentration of EC3-Pe in the lyotropic CLC mixture. Although the reflection peak wavelength of the lyotropic CLC was 480 nm at 30 °C when the polymer concentration was 54 wt% (Figure 4A, **Sample 1**), the reflection peak wavelength shifted to 533 nm and 663 nm when measured at the same temperature accompanied by the decrease in polymer concentration to 52 wt% or 50 wt%, respectively (Figure 4A, **Samples 2** and **3**). This plausibly happens due to the decrease in the helical twisting power of CLC or the expansion of its helical pitch length. Similarly, the red-shift of reflection peak with the decrease in polymer concentration was also observed for EC3-Pe_0.29_ (Figure 4B, **Samples 4**–**6**) and EC3-Pe_0.50_ (Figure 4C, **Samples 7**–**9**). These results fit with our expectations, which were based on the similar red-shift of the reflection peak wavelength observed for lyotropic CLCs from aqueous solutions of pristine HPC [1,12] and for a pristine EC solution in AA [17]. In both HPC and EC systems, the reflection peaks were broadened at lower polymer concentrations. The phenomenon can be explained by the decrease in the helical twisting power.

When AA was used instead of MAA as a solvent of lyotropic CLCs, the reflection peak also shifted to the longer wavelength side as the polymer concentration decreased for EC3-Pe_0.12_ (Figure 4D, **Samples 10**–**12**) or EC3-Pe_0.29_ (Figure 4E, **Samples 13**–**15**). However, the reflection peak of EC3-Pe_0.50_ dissolved in AA at a polymer concentration of 65 wt% did not appear in its transmission spectrum in the visible wavelength range of 400–800 nm, which was the detectable range of the measurement instrument (Figure 4F, **Sample 16**). To investigate this phenomenon, we conducted POM observation of **Sample 16**. The POM image at 30 °C showed transmitted light under cross-Nicols, revealing the emergence of optical birefringence through liquid crystallinity (Appendix A) [17]. When the temperature was increased from 30 °C to 110 °C, blue light of ~400 nm was found to be reflected (Appendix A). According to the precedent by Nishio and co-workers, the reflection peak wavelength of AA solutions of esterified EC derivatives shifts to the shorter wavelength upon heating [31]. Additionally, the reflection peak of lyotropic CLCs of EC3-Pe_0.12_ and EC3-Pe_0.29_ in AA (**Samples 10**–**15**) also shifted to the shorter wavelength side upon heating (Table 3). These results contextualized why the reflection peak appeared within the near-infrared wavelength region at 30 °C for **Sample 16**. Therefore, further research was conducted using the lyotropic CLC mixtures of EC3-Pe in MAA (**Samples 1**–**9**) to investigate the dependence of optical properties on *DS*_Pe_ value.

### 3.5. Reflection Peak Wavelength of EC3-Pe in MAA Dependence on DS_Pe_

Interestingly, we found that simply changing the *DS*_Pe_ value of EC-Pe gives rise to a significant difference in the shifting wavelength range of the reflection peak of its lyotropic CLC mixture in MAA upon conducting a stepwise heating process. In this study, lyotropic CLCs exhibiting blue, green, and red reflection colors at 30 °C were prepared from a series of EC3-Pe in MAA with different *DS*_Pe_ values (**Samples 1**–**9**). The changes in transmission spectra of these lyotropic CLCs upon heating are summarized in Appendix A. The sample temperature was increased from 30 °C at temperature intervals of 10 °C. The measurements were conducted in the temperature range at which the reflection peak of CLC appeared and in the wavelength range between 400 nm and 800 nm by considering the detectable wavelength limit of the CCD photodetector. The reflection peak wavelength at 30 °C, corresponding to the starting temperature for the measurements, was defined as *λ*_30 °C_ (Table 3, 5th column from left side). The reflection peak wavelength at the end of the measurements was denoted as *λ*_end_ (Table 3, 6th column).

Figure 5 plots the change in the reflection peak wavelength of lyotropic CLCs from EC3-Pe in MAA with a *DS*_Pe_ value of 0.12 (Figure 5A, **Samples 1**–**3**) to a *DS*_Pe_ of 0.29 (Figure 5B, **Samples 4**–**6**) and of 0.50 (Figure 5C, **Samples 7**–**9**) upon heating from 30 °C in a stepwise manner. In Figure 5, the plots are color-coded, and each color represents the reflection of the lyotropic CLC at 30 °C. For example, red plots in Figure 5 are those where the reflection color of lyotropic CLC was red at 30 °C.

In the case of **Samples 1**–**3**, the reflection peak wavelength of each lyotropic CLC continuously shifted to the shorter wavelength side upon heating (Figure 5A). When heated from 30 °C, the reflection peak wavelengths of **Samples 1**–**3** were changed from 480 nm to 402 nm (**Sample 1**), from 533 nm to 434 nm (**Sample 2**), and from 663 nm to 543 nm (**Sample 3**). Such changes in the reflection peak wavelength were especially apparent for **Samples 2** and **3** as their Bragg reflection color drastically changed upon heating (Figure 5A, images). These results were in good agreement with many precedents on CLCs from pristine EC or esterified EC derivatives [11,18,31]. Regardless of the difference in *λ*_30°C_, the reflection peaks of each lyotropic CLC disappeared when heated above 60 °C. Because the isotropic phase transition temperature (*T*_i_) of **Sample 3** was determined to be 44–68 °C through POM observation, we considered that such disappearance of the Bragg reflection color may be due to the transition of molecular orientation from liquid crystal phase to isotopic phase under heating. In order to quantitatively discuss the *DS*_Pe_ dependence of the wavelength shift of reflection peak upon heating, the measurement temperature range must be normalized. Therefore, the following equation was adopted to calculate the average wavelength shift range of reflection peak for every 10 °C, corresponding to the value of *λ*_shift_/10 °C.
(3)$$\lambda_{\mathrm{shift}}/10\;{}^{\circ}C=\left\{\left(\lambda_{30\;{}^{\circ}C}-\lambda_{\mathrm{end}})/(T_{\mathrm{end}}-30\right)\right\}\times 10$$

In Equation (3), *T*_end_ is the highest temperature of the measurement. The calculations show that at the *DS*_Pe_ value of 0.12, the *λ*_shift_/10 °C values for **Sample 1**, **Sample 2**, and **Sample 3** are estimated to be 25 nm, 33 nm, and 60 nm, respectively.

For the lyotropic CLCs from EC3-Pe in MAA with a *DS*_Pe_ value of 0.29 (**Samples 4**–**6**), the temperature dependence of the reflection peak wavelength was smaller than those from EC3-Pe in MAA with a *DS*_Pe_ of 0.12 (Figure 5B). The reflection peak shifted from 455 nm to 415 nm (**Sample 4**), from 520 nm to 418 nm (**Sample 5**), and from 705 nm to 497 nm (**Sample 6**) upon heating from 30 °C to *T*_end_. In particular, the change in the reflection peak wavelength was not so apparent for **Samples 4** and **5** in the temperature range of 30 °C to 60 °C (Figure 5B, green and blue triangles). Furthermore, the *λ*_shift_/10 °C of **Samples 4**–**6** were 7 nm, 16 nm, and 35 nm, respectively. These values were much smaller than those of **Samples 1**–**3**, as summarized in Table 3. Therefore, it was found that the *λ*_shift_/10 °C of lyotropic CLCs can be reduced by utilizing EC3-Pe with a relatively higher *DS*_Pe_. Notably, the measured temperature range was expanded to 30–90 °C. The mechanism for the expansion of temperature range for visible light reflection was the increase in *T*_i_ of these lyotropic CLCs. For example, the *T*_i_ of **Sample 6** was 90–98 °C, which was ~40 °C higher than that of **Sample 3**. Thus, the disappearance of the reflection peak upon heating can be ascribed to the transition from the CLC phase to isotropic phase.

The temperature dependence of the reflection peak wavelength of lyotropic CLC was smallest when using EC3-Pe in MAA with a *DS*_Pe_ vale of 0.50, that is, the completely etherified EC derivative (Figure 5C). The reflection peak wavelength only changed from 426 nm to 436 nm (**Sample 7**), from 524 nm to 495 nm (**Sample 8**), and from 639 nm to 545 nm (**Sample 9**) upon heating from 30 °C to *T*_end_. In addition, the values of *λ*_shift_/10 °C were 1 nm (**Sample 7**), 4 nm (**Sample 8**), and 12 nm (**Sample 9**). In these cases, the reflection peak disappeared on heating to 110 °C, irrespective of the reflection color at the starting temperature of the measurement. Considering that the *T*_i_ of **Sample 9** emerges above 155 °C, it was considered that the disappearance of reflection peak is due to light scattering caused by bubbles in the liquid crystal cell generated by heating above 110 °C, probably because of the boiling point of the MAA (160 °C) or the expulsion of MAA encapsulated in the EC-Pe polymer chain, which will be explained in the following section. As mentioned above, the *λ*_shift_/10 °C values of **Samples 7**–**9** were much smaller than those of **Samples 1–6**. The Supporting Video file shows the difference in Bragg reflection color change of **Samples 2** and **8** upon heating in the temperature range of 30–110 °C at 20 °C/min (Appendix A). After the reflection color of **Sample 2** was green at 30 °C, it changed to blue at ~50 °C and finally disappeared at ~85 °C. The disappearance of Bragg reflection color is ascribed to the reflection of ultraviolet light. In contrast, we found a unique reflection behavior whereby the green reflection color of **Sample 8** was hardly changed upon heating. Indeed, the reflection peak of **Sample 8** was almost maintained at ~520 nm even when heating (Figure 5C, green plots). Such a peculiar thermal stability of the Bragg reflection color of **Sample 8** expands the potential for CLCs from EC derivatives to be applied to new photonic devices with eco-friendliness.

During systematic studies, we found a tendency for the *T*_i_ points of lyotropic CLCs to rise when utilizing EC3-Pe with higher *DS*_Pe_. For instance, the *T*_i_ points of lyotropic CLCs showing a red reflection color at 30 °C were found to be 44–68 °C (**Sample 3**), 90–98 °C (**Sample 6**), and above 155 °C (**Sample 9**). Considering that each lyotropic CLC consisted of EC-Pe with the *DS*_Pe_ values of 0.12, 0.29, and 0.50, as given in Table 3, it was suggested that the *T*_i_ point of CLC is greatly dependent on the *DS*_Pe_ of EC3-Pe. This stems from the difference in MAA concentration of each lyotropic CLC, because the lyotropic CLC phase is more likely to be retained with the reduction in the number of solvent molecules that can flow upon heating.

The viscoelastic behavior of the lyotropic CLC of EC3-Pe dissolved in MAA was investigated to unravel the mechanism for the difference in the amount of reflection peak wavelength shift upon heating depending on the *DS*_Pe_ of EC3-Pe. Appendix A shows the angular frequency (*ω*) dependence of the storage modulus (*G*′) and the loss modulus (*G*″) of **Samples 1**, **4**, and **7** (Appendix A, red plots for **Sample 1**, blue plots for **Sample 4**, and black plots for **Sample 7**). To obtain reproducible data, pre-treatment was performed before each measurement based on previous studies [30,32]. As shown in Appendix A, *G*′ was greater than *G*″ in the entire *ω* range of 0.1–100 rad/s, implying the gel-like behavior of lyotropic CLCs. Although this behavior is not consistent with our previous rheological studies on the CLCs from HPC derivatives [30,32,33], these results are reasonable if we consider that the hydrogen bonding between MAA prohibits flow behavior. Additionally, the inflection points of *G*″ for the EC3-Pe solution in MAA were found at 6.3 rad/s for **Sample 1**, 2.5 rad/s for **Sample 4**, and 1.0 rad/s for **Sample 7**. These inflection points were more apparent when the loss tangent of *G*″/*G*′ was plotted against *ω* (Appendix A). As shown in Appendix A, the inflection point appeared shifted toward the lower *ω* side as *DS*_Pe_ increased. Based on the precedent from the literature [33], it is plausible that the inflection points appeared in the *ω* dependence of *G*″ are inseparably related to the tilting motion of the helical axis of the CLC, which is likely to be hindered with the increase in *DS*_Pe_ value as they shift to the lower *ω* side. These results lead us to consider that the reflection peak wavelength of the lyotropic CLC of EC3-Pe with a higher *DS*_Pe_ is maintained upon heating because of the restriction of molecular motion.

However, even when using EC3-Pe in MAA with the *DS*_Pe_ value of 0.50, the reflection peak wavelength of **Sample 9** shifted to 94 nm when heating from 30 °C to the *T*_end_ point (Figure 5C, red plots). This could be due to the expulsion of solvent molecules to the outside of CLC layers when heating. To prove this hypothesis, the changes in diameter of the lyotropic CLC sandwiched between glass plates were evaluated upon heating. The relative change ratio in the diameter (*d*_change_) of the lyotropic CLC was calculated as follows:(4)$$d_{\mathrm{change}}=d_{110\;{}^{\circ}C}/d_{30\;{}^{\circ}C}$$
where *d*_30°C_ and *d*_110°C_ are the diameter of the CLC at 30 °C and 110 °C, respectively. The *d*_change_ values of **Samples 7**–**9** were estimated to be 1.00, 1.05, and 0.94, respectively. It should be noted that only in **Sample 9** is *d*_change_ smaller than 1.0. This is because the solvent molecules inserted into the polymer networks of EC3-Pe are released from the CLC helical pitches due to heating. As discussed earlier in Section 3.4, the helical twisting power of **Sample 9** was weaker than those of **Samples 7** and **8**. In addition, the motility of liquid crystal molecules is enhanced by heating, which prohibits their orientation. Therefore, it can be considered that solvent molecules inserted into the polymer chain are more likely to be ejected in **Sample 9** rather than those in **Samples 7** and **8**. The solvent molecules inserted between the CLC layers are ejected, which shortens the helical pitch length of CLC, resulting in a shorter wavelength shift of the reflection peak wavelength.

## 4. Conclusions

In this study, we successfully prepared completely etherified EC derivatives through the Williamson ether synthesis. The EC derivatives with pentyl ether groups showed a lyotropic CLC phase with visible Bragg reflection when we dissolved them in MAA. The sharp reflection peak appeared only when utilizing pristine EC with a relatively low molecular weight because the lower viscosity of CLC enabled its better orientation. Moreover, the reflection peak wavelength of CLC could be maintained upon heating by utilizing the completely etherified EC derivatives. This can be ascribed to the restriction of the molecular movement of the liquid crystal structure, which was greatly affected by the differences in the substitution degrees of EC derivatives. These experimental results support us not only to understand the fundamental physical properties of CLCs from EC derivatives but also to create environmentally friendly CLC photonic devices using cellulose derivatives, which may help us to achieve a sustainable society.

## Figures and Tables

**Figure 1 polymers-16-00401-f001:**
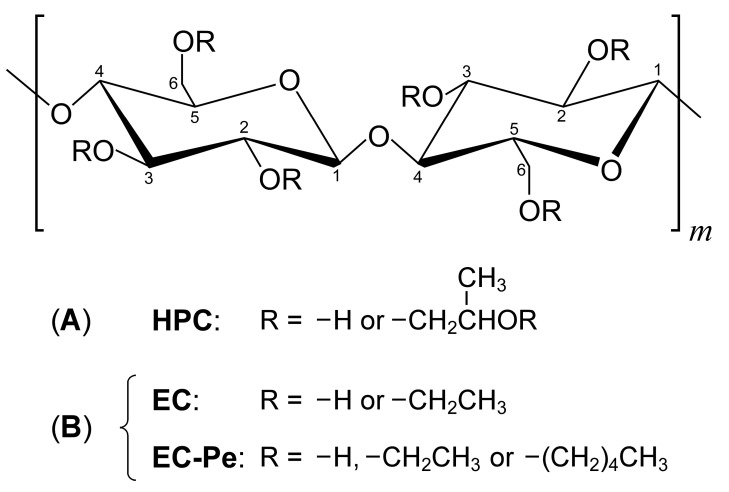
(**A**) Chemical structure of hydroxypropyl cellulose (HPC). (**B**) Chemical structures of ethyl cellulose (EC) and its derivative tethering pentyl ether side chains (EC-Pe) synthesized in this study.

**Figure 2 polymers-16-00401-f002:**
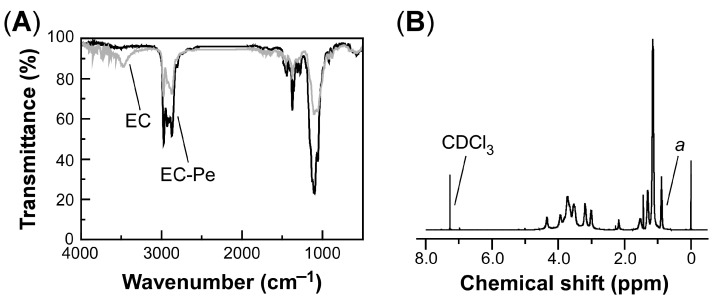
(**A**) FT-IR spectra of pristine EC (gray line) and EC3-Pe_0.5_ (black line). (**B**) ^1^H-NMR spectrum of EC3-Pe_0.5_ in CDCl_3_. The peak *a* is assigned to terminal methyl groups in the pentyl ether side chains of EC3-Pe_0.5_.

**Figure 3 polymers-16-00401-f003:**
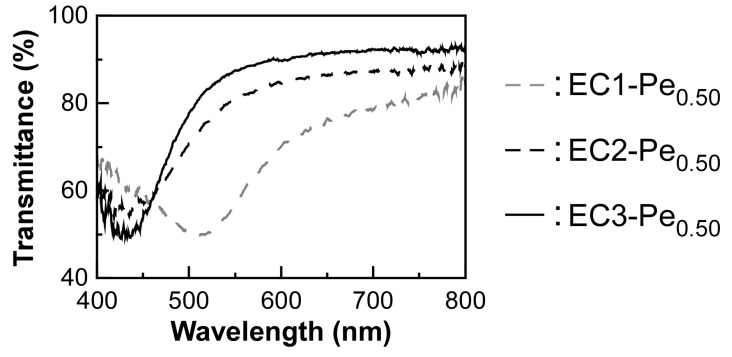
Transmission spectra of the CLC cells of EC1-Pe_0.50_, EC2-Pe_0.50_, and EC3-Pe_0.50_ at 30 °C.

**Figure 4 polymers-16-00401-f004:**
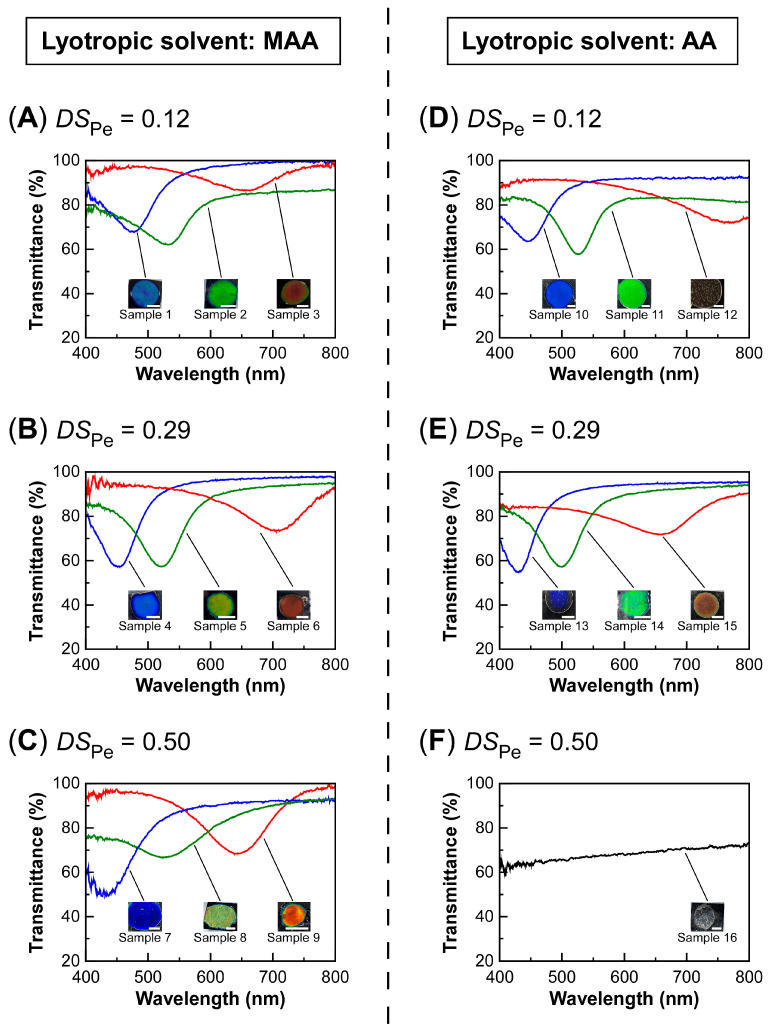
Changes in transmission spectra of CLC cells of lyotropic EC3-Pe in MAA or EC3-Pe in AA mixtures as a function of the concentration of EC3-Pe. The measurement temperature is 30 °C. The insets present the reflection images of CLC cells at the same temperature with white scale bars of 5.0 mm. (**A**) EC3-Pe_0.12_ in MAA, **Samples 1**–**3**. (**B**) EC3-Pe_0.29_ in MAA, **Samples 4**–**6**. (**C**) EC3-Pe_0.50_ in MAA, **Samples 7**–**9**. (**D**) EC3-Pe_0.12_ in AA, **Samples 10**–**12**. (**E**) EC3-Pe_0.29_ in AA, **Samples 13**–**15**. (**F**) EC3-Pe_0.50_ in AA, **Sample 16**.

**Figure 5 polymers-16-00401-f005:**
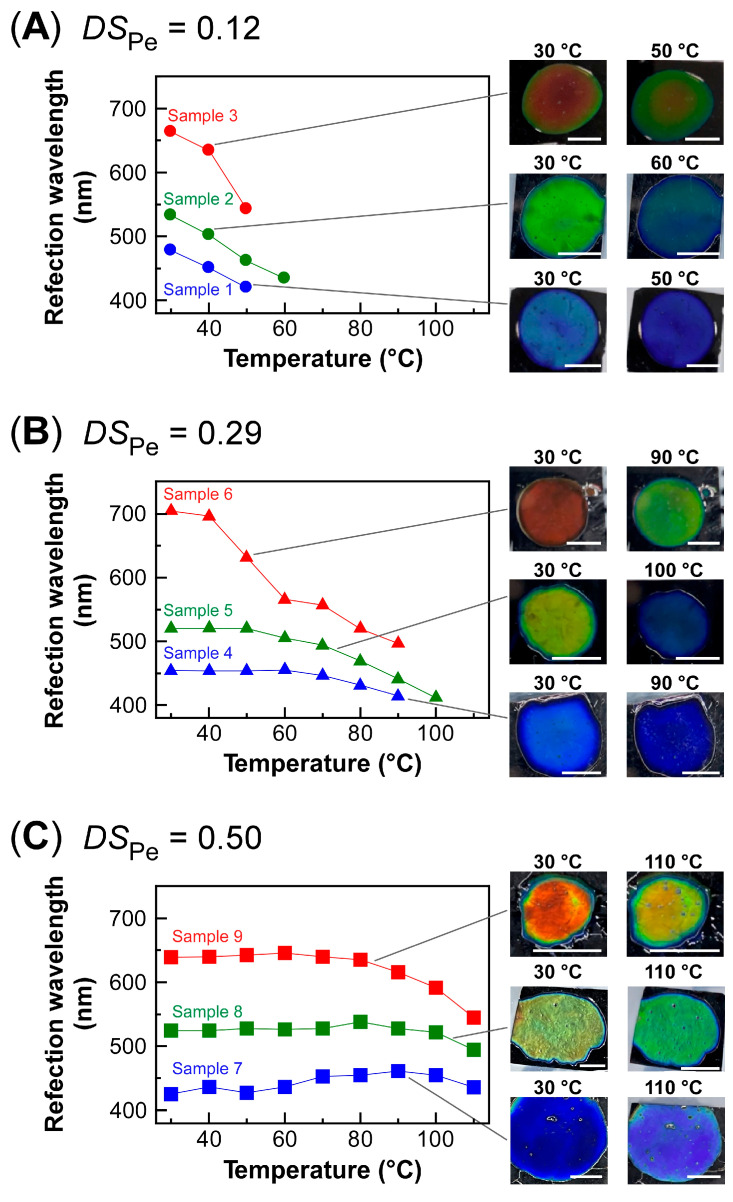
Temperature dependences of the reflection peak wavelength of lyotropic CLC from EC3-Pe with different *DS*_Pe_. (**A**) EC3-Pe_0.12_ in MAA, **Samples 1**–**3**. (**B**) EC3-Pe_0.29_ in MAA, **Samples 4**–**6**. (**C**) EC3-Pe_0.50_ in MAA, **Samples 7**–**9**. The insets are the reflection images of each lyotropic CLC at indicated temperatures with white scale bars of 5.0 mm. The plots are color-coded, and each color represents the reflection color of the lyotropic CLC at 30 °C.

**Table 1 polymers-16-00401-t001:** Synthesis conditions of EC-Pe and *DS*_Pe_ values.

Sample Code	Solvent	NaOH Concentration (g/mL)	Viscosity of Pristine EC (mPa·s)	*DS*_Pe_ *^a^*
EC1-Pe_0.15_	NMP	0.03	90–110	0.15
EC1-Pe_0.34_	DMAc	0.03	90–110	0.34
EC1-Pe_0.36_	DMAc	0.05	90–110	0.36
EC1-Pe_0.50_	DMAc	0.10	90–110	0.50
EC2-Pe_0.50_	DMAc	0.10	45–55	0.50
EC3-Pe_0.50_	DMAc	0.10	9–11	0.50
EC3-Pe_0.12_	DMAc	0.02	9–11	0.12
EC3-Pe_0.29_	DMAc	0.03	9–11	0.29

*^a^* The maximum *DS*_Pe_ value of 0.50 represents the completely etherified EC derivative with pentyl ether side chains.

**Table 2 polymers-16-00401-t002:** SEC results of pristine EC and EC-Pe.

Sample Code	*M*_n_ × 10^−5^	*M*_w_ × 10^−6^	*M*_w_/*M*_n_
EC1	6.74	4.29	6.36
EC2	5.02	2.43	4.84
EC3	2.58	0.762	2.95
EC1-Pe_0.15_	4.88	2.51	5.15
EC1-Pe_0.34_	4.92	2.74	5.58
EC1-Pe_0.36_	4.54	2.08	4.59
EC1-Pe_0.50_	5.02	2.28	4.53
EC2-Pe_0.50_	4.28	1.57	3.68
EC3-Pe_0.50_	2.61	0.770	2.95
EC3-Pe_0.12_	2.15	0.742	3.45
EC3-Pe_0.29_	2.16	0.666	3.09

**Table 3 polymers-16-00401-t003:** Sample definition of lyotropic CLCs and their CLC properties relevant for changing the temperature.

Sample	*DS* _Pe_	Solvent	Polymer Conc. (wt%)* ^a^*	*λ*_30 °C_ (nm) *^b^*	*λ*_end_ (nm) *^c^*	*λ*_shift_/10 °C (nm) *^d^*	*d*_change_ *^e^*
1	0.12	MAA	54	480	402	25	0.92
2	0.12	MAA	52	533	434	33	0.97
3	0.12	MAA	50	663	543	60	0.90
4	0.29	MAA	60	455	415	7	0.96
5	0.29	MAA	58	520	418	16	0.96
6	0.29	MAA	56	705	497	35	0.93
7	0.50	MAA	69	426	436	1	1.00
8	0.50	MAA	67	524	495	4	1.05
9	0.50	MAA	63	639	545	12	0.94
10	0.12	AA	69	440	428	– *^§^*	– *^§^*
11	0.12	AA	56	526	455	– *^§^*	– *^§^*
12	0.12	AA	53	771	486	– *^§^*	– *^§^*
13	0.29	AA	65	429	415	– *^§^*	– *^§^*
14	0.29	AA	62	500	410	– *^§^*	– *^§^*
15	0.29	AA	60	656	417	– *^§^*	– *^§^*
16	0.50	AA	65	– *	– *	– *^§^*	– *^§^*

*^a^* Weight concentration. *^b^* Reflection peak wavelength at 30 °C. *^c^* Reflection peak wavelength at the end of the measurement. *^d^* The average wavelength shift of the reflection peak for every 10 °C. *^e^* The relative change ratio in the diameter of the lyotropic CLC. *** The values were not known because the reflection peaks could not be measured. *^§^* Not measured.

## Data Availability

Data are contained within the article.

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
