# Peer review of "Cholesteric Liquid Crystals with Thermally Stable Reflection Color from Mixtures of Completely Etherified Ethyl Cellulose Derivative and Methacrylic Acid"

_polymers, 2024, doi:10.3390/polym16030401_

Round 1

Reviewer 1 Report

Comments and Suggestions for Authors

The manuscript titled “Cholesteric Liquid Crystals with Thermally Stable Reflection Color from Mixtures of Completely Etherified Ethyl Cellulose Derivative and Methacrylic Acid” reports an important research area. The paper is well written and will contribute significantly to the existing literature. I recommend its publication after major revision with respect to the following points.

1.     The abstract should be updated with characterization as well.

2.     Line 62; “The reflection peak wavelength of CLC is significantly dependent on the concentration of HPC or EC solutions, leading to their wide range of potential applications as concentration indicators, reflective color displays, full color recording media, tunable lasers and so.” The reflection peak or λmax is a characteristic of a compound. It does not change with concentration of respective compound. The authors should re-visit the given sentence.

3.     Similarly, line 70; “….the reflection peak of CLC can be controlled by temperature in either case.” As the reflection peak depends on molecular structure of a compound, it seems un-correct to say that reflection peak can be controlled by temperature.

4.     Line 122; Unit of molecular weight should be added.

5.     Line 148; NaOH has been used in powder form. According to literature, NaOH is always used in the form of solution. Is there any specific reason for using NaOH in solid form??

6.     Line 202; Only one peak has been discussed in FTIR in range of 3000-3600 cm-1. However, there are other peaks as well in FTIR. All peaks must be indexed.

7.     The data given in Table 2 is not correct. According to the given format, the Mn for sample 1 is 6.74 x 10-5 and so on. For a value of 6.74 x 105, the correct format is Mn ( × 10-5) (top of Table 2).

8.     Figure 5; what is the reason for temperature dependance of reflection wavelength? It should be explained in discussion.

9.     The whole manuscript must bee screened for English mistakes.

Comments on the Quality of English Language

Minor English editing is needed.

Author Response

Reviewer 1
============================================================
To Reviewer #1
Thank you very much for your kind review and generous encouragement to our report. Our manuscript has been revised according to your comments. The revised points have been highlighted in RED color on this revised manuscript.
============================================================

Comment #1
The abstract should be updated with characterization as well.
Response #1
Based on your suggestion, we revised the sentences at Lines 15–17 in Page 1 as follows. 
The degree of substitution with pentyl ether group (DSPe), confirmed by 1H-NMR spectroscopic measurements, was controlled by solvent and the base concentration in this synthesis.

Comment #2
Line 62; “The reflection peak wavelength of CLC is significantly dependent on the concentration of HPC or EC solutions, leading to their wide range of potential applications as concentration indicators, reflective color displays, full color recording media, tunable lasers and so.” The reflection peak or MAX is a characteristic of a compound. It does not change with concentration of respective compound. The authors should re-visit the given sentence.
Response #2
As shown in Refs. [13] and [17], it is well-known that the reflection peak of CLCs from the solutions of HPC or EC can be controlled by their concentration, arising from the lyotropic cholesteric liquid crystal property. Since the reflection color can be controlled simply by changing the concentration, we believe that applications described in the main text can be expected.

Comment #3
Similarly, line 70; “….the reflection peak of CLC can be controlled by temperature in either case.” As the reflection peak depends on molecular structure of a compound, it seems un-correct to say that reflection peak can be controlled by temperature.
Response #3
As you pointed out, the reflection peak depends on the molecular structure of the compound. However, as shown in Refs. [25] and [27], the reflection peak wavelength shifts to the longer wavelength side upon heating process for the HPC derivatives with esterified or etherified side chains. Such thermally induced shifting behavior of the reflection peak originates from the thermotropic liquid crystal characteristic. Based on your suggestion, we added a following text at Lines 72–73 in Page 2.
Moreover, the reflection peak of CLCs from HPC derivatives generally shift to the longer wavelength side upon heating process.

Comment #4
Line 122; Unit of molecular weight should be added.
Response #4
The molecular weight doesn’t have a unit because it is defined as a relative mass based on the mass of carbon atom. Likewise, this definition is the same for atomic weight.

Comment #5
Line 148; NaOH has been used in powder form. According to literature, NaOH is always used in the form of solution. Is there any specific reason for using NaOH in solid form??
Response #5
The first reason is to minimize the contamination of water because Williamson ether reaction was used to obtain the target product. Moreover, it is necessary to dissolve EC and 1-bromopentane uniformly to obtain completely etherified EC derivatives. Therefore, NaOH must be added in powder form to the solution of EC in DMAc.

Comment #6
Line 202; Only one peak has been discussed in FTIR in range of 3000-3600 cm−1. However, there are other peaks as well in FTIR. All peaks must be indexed.
Response #6
Based on your suggestion, we added the explanation about the peaks at 2840–3000 cm−1of FT-IR. However, it is not possible to assign all peaks especially in the fingerprint region because of the complexity of the chemical structure of HPC derivatives. Based on your suggestion, we have revised the sentence at Lines 210–212 in Page 5 as follows.
Furthermore, the intense peak from the C-H stretching vibration at 2840–3000 cm−1 became stronger, suggesting that etherification has proceeded.

Comment #7
The data given in Table 2 is not correct. According to the given format, the Mn for sample 1 is 6.74 × 10-5 and so on. For a value of 6.74 × 105, the correct format is Mn ( × 10-5) (top of Table 2).
Response #7
Thank you for pointing out our mistake. We have corrected the part you pointed out.

Comment #8
Figure 5; what is the reason for temperature dependance of reflection wavelength? It should be explained in discussion.
Response #8 
As described in the Introduction part, the helical pitch length of CLC (p) greatly depends on the temperature as a typical characteristic of the thermotropic liquid crystal. Therefore, the reflection peak wavelength of CLC also changes by temperature.
In the case of our CLCs from EC derivatives dissolved in MAA, the angular frequency dependence of the storage modulus and the loss modulus suggest that the tilting motion of the helical axis of the CLC depends on DSPe. This implied that the molecular helix of CLC from EC-Pe with DSPe = 0.5 are unlikely to move, which also prohibited the modulation of p upon heating process. The discussion about this peculiar property is available at Lines 469–489 in Page 14.

Comment #9
The whole manuscript must be screened for English mistakes.
Response #9
The revised manuscript has been checked by a native speaker of English. Therefore, we believe that the English mistakes are removed from this revised manuscript as much as possible.

Reviewer 2 Report

Comments and Suggestions for Authors

The submitted report collects recent advances on cellulose derivatives that exhibit cholesteric liquid crystal phase with visible light reflection.

The manuscript is well structured, well written and the references are appropriate.

My opinion is to accept the submission but I ask the authors: 

- was a second procedure carried out to evaluate the actual reproducibility of the data? Could be reported...

- is it possible to enlarge the images in figure 5? The figure can be larger in size to improve the quality

Author Response

Reviewer 2

============================================================

To Reviewer #2

Thank you very much for your kind review and highly encouraging comment. Our manuscript has been revised according to your comments. The revised points have been highlighted in RED color on this revised manuscript.

============================================================

Comment #1

The submitted report collects recent advances on cellulose derivatives that exhibit cholesteric liquid crystal phase with visible light reflection. The manuscript is well structured, well written and the references are appropriate. My opinion is to accept the submission.

Response #1

We appreciate for your highly encouraging comment to our manuscript.

Comment #2

- was a second procedure carried out to evaluate the actual reproducibility of the data? Could be reported...

Response #2

All measurements are performed at least twice to check the reproducibility. As an example, Figure R1 shows the changes in the reflection peak wavelength upon heating process of Sample 3. The red plots are the result of the first-round measurement described in the main text, and the black plots are the result of the second-round measurement, which were almost identical to those of the first-round measurement. Therefore, we believe that the experimental results include high reproducibility.

Please check the Figure R1 in attached PDF file.

Figure R1. Temperature dependences of the reflection peak wavelength of Sample 3. The results of the first-round and second-round measurements are shown as red and black plots, respectively. Note that the plots of the first-round measurement are the same as the red plots in Figure 5A in the main text.

Comment #3

- is it possible to enlarge the images in figure 5? The figure can be larger in size to improve the quality.

Response #3

Based on your suggestion, we have enlarged the pictures shown as the insets in Figure 5 of this revised manuscript.

Round 2

Reviewer 1 Report

Comments and Suggestions for Authors

The manuscript has been revised accordingly.